# Threatening language detection from Urdu data with deep sequential model

Ashraf Ullah[1]*, Khair Ullah Khan[1], Aurangzeb Khan[1], Sheikh Tahir Bakhsh[2], Atta Ur Rahman[3], Sajida Akbar[1], Bibi Saqia[1]

1 Department of Computer Science, University of Science & Technology Bannu, Bannu, Khyber Pakhtunkhwa, Pakistan, 2 Cardiff School of Technologies, Cardiff Metropolitan University, Cardiff, United Kingdom, 3 Riphah institute of system engineering (RISE), Riphah International University, Islamabad, Pakistan

* ashrafbth@gmail.com

## Abstract

The Urdu language is spoken and written on different social media platforms like Twitter, WhatsApp, Facebook, and YouTube. However, due to the lack of Urdu Language Processing (ULP) libraries, it is quite challenging to identify threats from textual and sequential data on the social media provided in Urdu. Therefore, it is required to preprocess the Urdu data as efficiently as English by creating different stemming and data cleaning libraries for Urdu data. Different lexical and machine learning-based techniques are introduced in the literature, but all of these are limited to the unavailability of online Urdu vocabulary. This research has introduced Urdu language vocabulary, including a stop words list and a stemming dictionary to preprocess Urdu data as efficiently as English. This reduced the input size of the Urdu language sentences and removed redundant and noisy information. Finally, a deep sequential model based on Long Short-Term Memory (LSTM) units is trained on the efficiently preprocessed, evaluated, and tested. Our proposed methodology resulted in good prediction performance, i.e., an accuracy of 82%, which is greater than the existing methods.

## 1. Introduction

The definition of threat, according to Twitter, is a statement to impose serious physical harm or an intent to kill persons or an entire group. A threat is defined as a statement expressing severe harm, either bodily or in another form. For instance,"Keep your mouth shut, or you will be seen red." The word"red" can be depicted as a threat to injure someone or either kill someone in the worst case. This kind of remark is thus considered a vile aspersion. Twitter has taken some initiatives to stop the spread of threatening remarks on the platform. For instance timeout feature is used to suspend an account for hours that are found abusive. Even more, efforts are required to detect threatening and offensive content in different languages since abusive language is still prevalent in social media.

Social networks have become essential parts of our daily routine with the emerging communication technology and the Internet as the toll of users of social media is rising abruptly. For

**Data Availability Statement:** Now we have made the dataset publically available on GitHub, and can be accessed on this link "https://github.com/ashraf8484/Augmented-Threatening-Language-Urdu-Dataset". The same link is also mentioned in

the revised Manuscript as a footnote at page number 8.

**Funding:** The source of funding is Cardiff Metropolitan University UK. The study benefitted significantly from the funder's integral involvement. They actively participating in experiments, and offering valuable guidance throughout the analysis process. Their support significantly contributed to the overall success and robustness of the study.

**Competing interests:** The authors have declared that no competing interests exist.

instance, StatInvestor(https://statinvestor.com/) reported that from 2010 to 2021, the number of social media users had extended three times from 0.97 billion to 3.02 billion. Twitter has around 353 million active users per month, according to Statista (https://www.statista.com/), with 200 billion annual tweets. Twitter is used for writing, reading, and sharing short texts with a character limit of 280 per tweet.it is considered one of the most popular social media platforms. Such platforms have huge numbers of people with cultural, religious, ethnic, and linguistic diversity [1]. At the same time, freedom of speech is said to be restricted by censorship of free expression as Twitter has been used as a medium to commit cyber crimes like spamming, cyber-bullying, malware spreading, and phishing [2]. Furthermore, the challenging issues include encouragement to self-harm and sexual offending [3–6]. This can instigate threats against physical violence and gender-based violence [7]. GamerGate, a controversial online movement that emerged.

In 2014 within the gaming community, scandal can be an example where women were given death and rape threats by video game lovers on the Twitter platform.

Some users also use Twitter to threaten other people by posting threatening posts. This has initiated a growing body of research to investigate the use of threatening content in social media by detecting threatening content and abusive language [8, 9]. Given the anguish it causes social media users, further research in automatic threatening language detection is vitally important to solve this problem in large platforms like Twitter. While the technology for the automatic detection of threatening language is still in its early stage, Twitter has gone beyond only English to detect threatening language [10, 11]. Different studies investigated the detection of threatening languages automatically in different languages like German, Italian, Arabic, Indonesian, Bengali, and Dutch [7, 12–16]. These studies worked on linguistic features and lexical resources for the detection of threatening language automatically.

Urdu is a major Indo-Aryan language primarily spoken in Pakistan and India. It has a rich history and is known for its poetic and literary traditions. Urdu encompasses many emotional words and phrases that can effectively convey sentiments and emotions. This allows sentiment analysis models to capture and analyze nuanced sentiments expressed in Urdu text. Threats detection from Urdu poses a significant challenge due to limited data availability, linguistic diversity, and regional variations within Urdu-speaking communities. ML and DL models require diverse datasets for training to address these variations and accurately detect depressive sentiments effectively. The proliferation of Urdu language usage across various social media platforms like Twitter, WhatsApp, Facebook, and YouTube presents a significant challenge due to the absence of robust Urdu Language Processing (ULP) libraries. The lack of adequate resources hampers the identification and mitigation of potential threats concealed within textual and sequential data shared in Urdu.

Current literature underscores the limitations imposed by the absence of online Urdu vocabulary, constraining the applicability of lexical and machine learning-based techniques in effectively processing Urdu data. Existing methodologies fall short in efficiently preprocessing Urdu content akin to the processing capabilities available for English.

In response to this pressing need, this research endeavors to bridge the gap by introducing a comprehensive Urdu language vocabulary, featuring a meticulously curated stop words list and a dedicated stemming dictionary. The goal is to establish an efficient preprocessing framework for Urdu data, aimed at reducing input sentence size and eliminating redundant and noisy information.

Moreover, the study proposes the utilization of a deep sequential model leveraging Long Short-Term Memory (LSTM) units. This model is trained on the meticulously preprocessed Urdu data, subsequently evaluated and tested to assess its performance.

The primary objective of this research is to ascertain the effectiveness of the proposed methodology in enhancing predictive performance. The validation results demonstrate a promising accuracy rate of 82%, surpassing the efficacy of existing methodologies.

There are a few key flaws and differences in Urdu language processing and social media threat identification that need to be taken into consideration. First off, even though efforts have been made to identify language that poses a threat in a variety of languages, including Urdu, the preponderance of English-centric approaches leaves a significant gap in addressing the particular complexities and subtleties present in Urdu linguistic structures and cultural contexts, impeding the creation of effective threat detection systems. Second, the lack of extensive libraries for Urdu language processing makes it more difficult to identify risks in Urdu material on social media, which restricts the use of cutting-edge machine learning methods. Furthermore, as vocabulary and syntax variances present challenges to reliable threat recognition, geographical variations and various linguistic characteristics among Urdu-speaking groups further complicate the creation of broadly applicable threat detection models. Moreover, a major bottleneck that compromises threat detection models' efficacy and generalizability is the lack of annotated datasets for training and assessment. In order to advance Urdu language processing and threat detection, these issues must be resolved. This calls for the creation of novel approaches to get around these restrictions and promote the advancement of more potent strategies for detecting and averting threats on a variety of social media platforms.

This research's scope includes solving the difficulties associated with processing Urdu language, especially when it comes to threat identification on social media. The study tries to improve threat detection model efficiency by providing a broad vocabulary in Urdu and suggesting sophisticated preprocessing methods. Furthermore, the use of deep sequential models, including Long Short-Term Memory (LSTM) units, emphasises how critical it is to advance Urdu language processing technology in order to better identify and predict threats. The present study effectively mitigates dangers in Urdu textual data across many social media platforms, so contributing to the wider goal of fostering safer online environments. It also closes significant gaps in existing approaches.

Therefore, the fundamental focus of this study is to address the challenges inherent in Urdu language processing by establishing a robust preprocessing framework and validating its efficacy through the implementation of advanced deep sequential models, thus contributing to improved threat identification and prediction in Urdu textual data across social media platforms.

## 1.1. Contributions

In this paper, we proposed to employ advanced augmentation techniques in order to address the challenge of limited data availability for Urdu data. Furthermore, we perform data preprocessing to remove the noise efficiently with the help of the proposed Urdu stop words list and stemming dictionary. We apply deeper data augmentation steps leading us towards better training of LSTMs [17] for threatening Tweets detection. Our model trained over this preprocessed data results in improved outcomes than the existing state-of-the-art methods [18, 19]. We summarize our research contribution as follows:

- We proposed to apply enhanced data augmentation steps.

- We generated Urdu stop words list and stemming dictionary.

- We efficiently performed preprocessing of data to remove the noise.

- Finally, the proposed LSTM architecture is trained for threatening language detection.

## 1.2. Paper arrangement

The rest of our paper is managed in the respective sections as follows; Section 2 describes the available state-of-the-art and some key contributions. Section 3 explains the proposed methodology and network architecture. Section 4 is about a brief discussion of the experimental setup. Section 5 presents and discusses the results and comparison with other methods. The paper is concluded in Section 6, with some future directions.

## 2. Literature review

Detection of threatening language is challenging work, specifically differentiating it from other derogatory content or even kind content where there are chances of some flapping words. Negative terms are commonly used for sarcasm or amusement. For example, Blood, Stab, Murder, Death, and kill are commonly used words. The Natural Language Processing (NLP) community is working for different social media platforms like Twitter, WhatsApp, Facebook, YouTube, Blogs, and Instagram to detect threatening language [2, 20–29]. Various studies counted upon chi-square feature selection and lexicon-based techniques are used for the automatic detection of threatening language. Furthermore, character n-grams [12, 21, 22, 30–32], word n-grams, and a combination of both are also used by many researchers [1, 12, 16, 22–24, 26, 33, 34].

Some studies also use machine learning techniques and multilingual datasets to detect threatening language automatically. For instance, various studies proposed the usage of Logistic Regression (LR) classifiers and Support Vector Machines (SVMs) to identify offensive speech in blogs, tweets, Reddit, articles, and Facebook [12, 16, 21, 23, 24, 26, 29, 31, 33]. Likewise, Naïve Bayes (NB) was utilized to identify derogatory remarks in News Groups and comments from YouTube. Whereas for detecting threatening language in Turkish Instagram content and tweets, Decision Tree(DT) had been used [1, 22, 29, 31, 35]. Furthermore, a single study has utilized K-nearest neighbors (KNN) on the datasets of Instagram posts, comments, or tweets in the Turkish language [29].

Deep Learning models are also used in the latest studies to detect threatening language. These techniques were also employed for depression diagnosis and provided the best results [36]. For instance, to detect threatening content on Facebook and Twitter in German and English, Convolutional Neural Networks (CNNs) are used [1, 6, 12, 13, 20–25, 27, 30, 31, 33, 35]. Those studies proved that CNNs surpassed other neural network-based models. Bert's model for improving news article categorization performance in Bangla was used. This explains why blog postings, newspaper articles, and social media posts are all gaining popularity among the large Bangla-speaking populace [37]. Deep Learning algorithms have been used to identify various forms of music in a few significant ways. Bengali music is considerably more engaging because of its substance and distinctiveness. In addition, there is still a lot to learn about using the DL technique in Bengali music. As a result, Bengali music genre categorization is a relatively recent topic of study in the field of Deep Learning [38]. Furthermore, in the Bengali language, the LSTM model, which is a Recurrent Neural Network architecture, was used by a few research studies to detect threatening language on Facebook [12, 23, 26, 31]. English Twitter content used Graph Convolutional Network and BiLSTM [21, 23, 30, 32]. Some other researchers used threatening language detection in languages like German, English, Turkish, Italian, Bengali, Danish, Arabic, Japanese, Spanish, Portuguese, and Indonesian [1, 6, 12, 13, 15, 20–26, 29–35, 39].

Recent research has focused on detecting threatening language with machine learning from Urdu data [18, 19]. The first ever threats labeled Urdu dataset is proposed in 2021 with only 3,564 Tweets [18]. Both studies have applied different ML and DL methods, including Multi-

**Table 1. Literature performed for threats detection on Twitter data.**

| Author | Language | Features | Model |
|--------|----------|----------|-------|
| [24] | English | unigram | SVM, CNN |
| [31] | English | word n-grams | SVM |
| [13] | German | embedding | CNN |
| [15] | Italian | BERT tokens | AlBERTo |
| [18] | Urdu | n-grams, fasText | SVM, MLP, CNN |
| [19] | Urdu | data cleaning, n-grams | LR, CNN, LSTM |

Layer Perceptron (MLP), SVM, Extra Tree (ET), and Bernoulli Naive Bayes (BNB) to construct a Logistic Regression (LR) classifier. The best accuracy reported by both authors is around 75%, which is good but improvable. The issue is that handi-craft feature selection methods are used for the detection due to the unavailability of Urdu preprocessing libraries. In contrast, our research proposed Urdu stop words list and stemming dictionary for efficient preprocessing and better classification. Table 1 provides brief information about the available research for threat detection in different languages. As the table shows, only two research articles have focused on threat detection from Urdu data.

## 3. Methodology

The proposed methodology that is followed in this research comprises data acquisition (gathering and augmentation), preprocessing (data cleaning, stemming, tokenization, vectorization, padding), and modeling along with analysis (LSTM training, validation, testing, and comparison) as depicted in Fig 1.

### 3.1. Data acquisition

**3.1.1. Dataset selection.** This research has selected a manually labeled, as threaten or non-threaten, Urdu dataset containing 3,564 tweets [18]. The total number of threatening sentences and non-threatening sentences is 1,782 each. The total number of tweets in each class is little for a DL model, which is increased with the help of a data augmentation technique called back-translation. Sample sentences with respective labels from the dataset are;

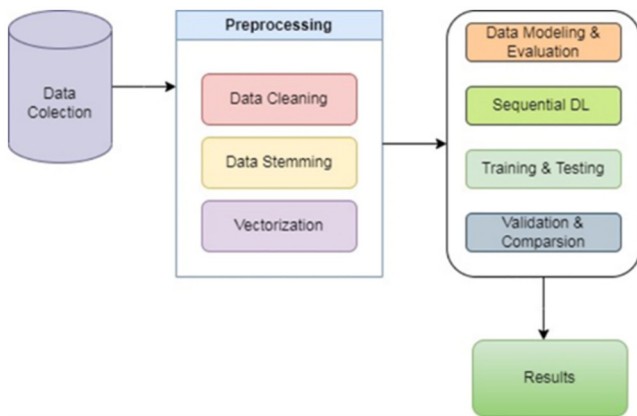

**Fig 1. Proposed methodology block diagram.**

بھگوڑی لیگ صرف اپنے مفاد تک ہے Non-Threaten

بکواس مت کرو Threaten

**3.1.2. Data augmentation.** Due to the limited labeled data of the Urdu language, data enhancement is required to resolve the overfitting and under-fitting of a deep learning model. If the quantity of data is less, then the chances of model under-fitting and overfitting are high [40]. To resolve the issue, the better solution is to acquire the data manually under the supervision of domain experts, but that is quite a hectic and time-consuming job. Another intelligent way of machine learning is to apply data augmentation techniques to enhance data [40]. In this research, we have used back-translation to enhance the number of samples. As the name suggests, back-translation uses translation from one language to another and then back-translates the translated text to the first language. Back-translation is done using Machine Translation Service (MTS), which depends on the random variations generated by the service. We have used Google Translation API using the Python platform. The total number of threatening sentences and non-threatening sentences is 7128 each. The dataset is publically available on GitHub (https://github.com/ashraf8484/Augmented-Threatening-Language-Urdu-Dataset).

بھگوڑی لیگ صرف اپنے مفاد تک:Original Tweet

Translated Tweet: The fugitive league is only up to its own interests

مفرور لیگ صرف اپنے فائدے کے لیے ہے :Back-translated

## 3.2. Preprocessing

Preprocessing is always considered one of the essential steps for any ML or DL model. Without preprocessing, the models may remain underfit due to noise.

**3.2.1. Data cleaning.** It is required to clean up the data to enhance the quality and improve any automated model's productivity [40]. Removal of special characters, white spaces, digits, and stop words is performed for data cleaning. Any less informative word frequently used in any language is considered a stop word for that language. Due to the unavailability of Urdu stop words, this research has introduced a list of 270 Urdu stop words generated from translated English stop words list and manual inspection.

## 3.3. Stemming

In linguistics, a single word can be presented in different forms depending on the Tense the word is used. For example, the word 'Read' can be used as 'Reads', 'Reading', or 'Readable.' The meaning of all the variants of a single word is the same, but computers and machines consider all the variants as different words. In Urdu, the variants are not limited to a few numbers. Table 2 shows a sample Urdu word variant. $V_n$ shows the possible variants for the root word 'پڑھ'. The total number of variants is more than 10. The number variants for the same root word in English are shown in the table, which is just 2. Another issue is that no Urdu

**Table 2. Sample stemming words.**

| Language | Root | $V_1$ | $V_2$ | $V_3$ | $V_4$ | $V_5$ | $V_6$ | $V_7$ | $V_8$ | $V_9$ | $V_{10}$ |
|---|---|---|---|---|---|---|---|---|---|---|---|
| Urdu | پڑھ | پڑھے | پڑھا | پڑھی | پڑھو | پڑھوں | پڑھتے | پڑھتے | پڑھیے | پڑھائے | پڑھائوں |
| English | read | reads | reading | | | | | | | | |

**Table 3. Vectorization Example.**

| Cleaned Tweet | | | | بھگوڑی لیگ مفاد تک |
|---|---|---|---|---|
| Vectorized Tweet | 31 | 124 | 24 | 17 |

stemming dictionary maps a word's different forms to its root word. In this research, a low-level Urdu stemming dictionary is proposed and created with the inspection of the literature expert. The total number of unique root words in the generated Urdu dictionary is approximately 1800.

## 3.4. Vectorization

Vectorization is a process of transforming textual data into numerical representations, as computers primarily comprehend numerical information. Various vectorization techniques are employed to facilitate this conversion, including indexing, count vectorizers, N-grams, and term frequencies. This research utilized the indexing method due to its simplicity and versatility across different grammatical styles. In the indexing method, each distinct word, known as a token, is extracted from the dataset. Subsequently, each unique token is assigned a unique numerical value, an index. For example, in Table 3 consider the word 'بھگوڑی' and the index number assigned to the word is '17'. Similarly, all the other unique words are given a unique index number. All the words are replaced with their respective indices. Consider the same sentence discussed in section 3.1.1 is shown in the vectorized form below. Any automated model prefers a fixed length of input vectors, while the resulting vectorized dataset has different lengths. The same padding is applied to the vectorized dataset to equalize the length of all tweets to 200.

## 3.5. Deep sequential model

This research used Long Short-Term Memory (LSTM) model [17] as a deep sequential model. An LSTM model takes care of Long-Term Memory (LTM) and Short-Term Memory (STM), as the name suggests. There are four different gates used in an LSTM unit. These gates are:

1. **Forget Gate**: The functionality of this gate is to forget less useful or infrequent information inside LTM.

2. **Learn Gate**: A current input (an event) and STM are merged to remember the recently learned information from STM and applied to that event.

3. **Remember Gate**: As the name suggests, the main functionality of this gate is to remember the previous information up to certain limits. LTM information which is not forgotten. An event and STM merged in this gate work as an updated version of LTM.

4. **Use Gate**: The functionality of Use Gate is to predict the current event's output using STM, LTM, and an Event.

A simple LSTM architecture is shown in Fig 2. The boxes in red, yellow, and blue are different parameters and activation functions of the above-mentioned gates.

## 4. Experimental setup

The proposed methodology was implemented on a Windows 10 operating system, utilizing Python 3.10 programming language on a core i7 desktop system with 16 GB of RAM. The LSTM model is trained with the help of the TensorFlow library. The performance measures

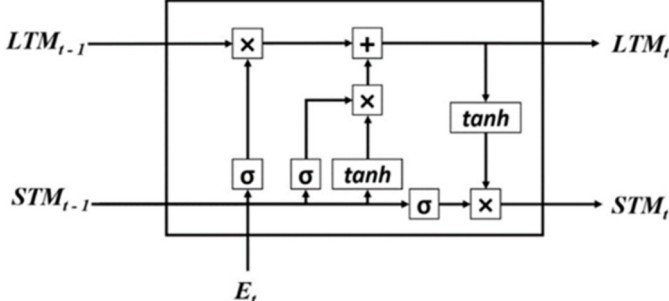

**Fig 2. An LSTM unit.**

used for evaluation and testing purposes include accuracy, f1-score, precision, and recall presented in 1, 3, 4, and 2, respectively, in terms of True Positive (TP), False Positive (FP), False Negative (FN), and True Negative (TN).

$$Accuracy = \frac{TP + TN}{TP + FP + FN + TN} \tag{1}$$

$$F1 - Score = \frac{2 * Precision * Recall}{Precision + Recall} \tag{2}$$

$$Precision = \frac{TP}{TP + FP} \tag{3}$$

$$Recall = \frac{TP}{TP + FN} \tag{4}$$

## 4.1. LSTM architecture

After successful preprocessing, the cleaned data is given to the LSTM model shown in Fig 3. The input layer has a shape of 200 followed by an embedding layer where random weights are given. The total number of parameters at this layer was 7150200. Then the model implemented an LSTM layer with an output shape of 256 neurons with a total of 467968 trainable

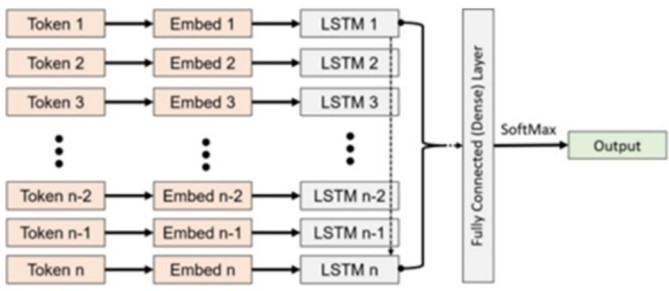

**Fig 3. Model architecture.**

parameters, followed by a dense layer with 32896 trainable parameters. The final output layers have a total of 2 labels because of the binary classification problem.

## 4.2. Hyper- and parameters configuration

For setting up parameters and hyper parameters of the LSTM model, Table 4 provides all the model configurations. Early stopping is beneficial to a model's convergence and hence avoided overfitting the LSTM model. The starting learning rate is finalized at 0.001. The early stopping was monitored using validation accuracy during training. Patience in stopping the training processes was set to 4.

# 5. Results and discussion

This section highlights an overview of the best results that are achieved for the parameters and hyper parameters given in Table 4. First, the dataset is split into train tests with a ratio of 80:20. From the training data, before passing it to the proposed model, we have split it further into an 80:20 ratio between the train and validation set. Moreover, callbacks from Table 4 are also applied based on validation loss while training the model to save the best weights.

## 5.1. Training

Fig 4a shows the graph which depicts the training accuracy versus validation accuracy.

X-axis represents the number of epochs, and the y-axis shows the accuracy value. The graph clearly shows that model validation accuracy increases as the number of epochs increases and hence the model is converging. There is no overfitting on the training data as the difference between training and validation performance measures is negligible. Also, Fig 4b shows that the model is learning with epochs and has stopped the training, after meeting the stopping criteria, at epoch number 15.

**Table 4. Parameters configuration.**

| Hyper-\Parameter | Configuration | |
|---|---|---|
| **Learning Rate** | Initial Value | 0.001 |
| | Nature | Adaptive |
| | Reduce factor | 0.1 |
| | Patience | 1 |
| | Minimum | 0.00001 |
| | Monitor | Val Loss |
| **Weights** | Initial weights Trainable | Random True |
| **Stopping Criteria** | Early Stopping | True |
| | Monitor | Val Loss |
| | Patience | 4 |
| **Training-Validation** | Optimizer | ADAM |
| | Loss | Categorical Cross |
| | Max Epochs | Entropy |
| | Batch Size | Inf |
| | | 512 |
| | Val Split | 20% |
| | Performance Metrics | Accuracy |
| | | Loss |
| | | Val Accuracy Val Loss |

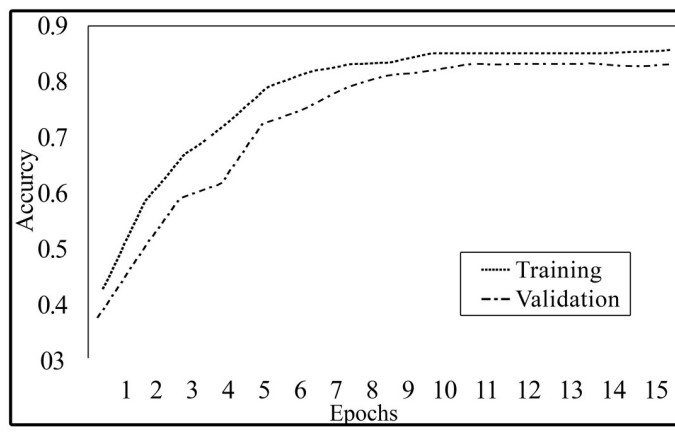
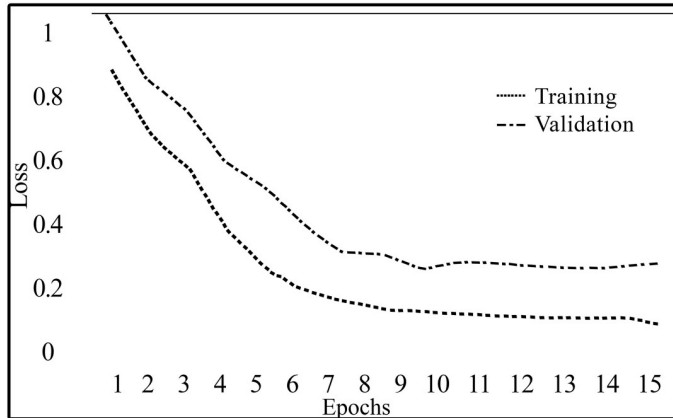

**(a)** Accuracy

**(b)** Loss

**Fig 4. LSTM model performance while training and validation.**

## 5.2. Testing

The model performance is checked for testing data after getting optimum weights for the training and validation set. The confusion matrix calculated on the testing data is given in Table 5, which shows TP, TN, FP, and FN which were 601, 568, 145, and 112, respectively. FN are the cases that were actually threats but detected as non-threats. Such sentences might be dangerous and should be reduced. It is noticed that the number of threats predicted by the proposed methodology is more than non-threats, which is a good practice.

Furthermore, the performance measures on the testing dataset are provided in Table 6. The best accuracy, precision, recall, and f1-score achieved by the proposed methodology are 81.97%, 82.04%, 81.97%, and 81.96%, respectively. All the performance measures are almost the same, which shows that the proposed methodology does not over-fit the LSTM architecture.

## 5.3. Comparison with literature

As discussed in section 2, only two papers have worked in detecting threatening language from Urdu data [18, 19]. This research has compared the overall methods and performance with the state-of-the-art methods in the literature. Some well-known ML methods, including

**Table 5. Confusion matrix on testing data.**

|  | Threat | Non-Threat |
|---|---|---|
| Threat | 601 | 112 |
| Non-Threat | 145 | 568 |

**Table 6. Comparison of classes.**

| Class | Precision | Recall | F1-Score |
|---|---|---|---|
| Threat | 80% | 84% | 82% |
| Non-Threat | 83% | 79% | 82% |

**Table 7. Methodology comparison between the proposed work and literature.**

| Author | Model | Tweets | Features Selection | Stop Words Removal | Stemming | Accuracy (%) |
|--------|-------|--------|--------------------|--------------------|----------|--------------|
| [18] | SVM, MLP, CNN | 3,564 | Yes | No | No | 72.5 |
| [19] | LR, CNN, LSTM | 3,564 | Yes | Yes | No | 74.01 |
| Temporary | LSTM | 3,564 | No | No | No | 65.25 |
| Proposed | LSTM | 7,128 | No | Yes | Yes | **81.96** |

SVM, LR, and MLP, are initially trained on the features selected from the proposed dataset [18]. This research has not used any preprocessing to detect better threats, which is not recommended [40]. Similarly, another research used features selection and removing stop words from the dataset but did not use any stemming process.

In contrast, the proposed research has let the LSTM model automatically select features and tune the parameters along with the stop words removal and stemming the words to the respective roots. Table 7 briefly compares the proposed method and the available research to the best of our knowledge. We have further validated our proposed flow of preprocessing steps by performing experiments without data preprocessing, and the accuracy is compared. Fig 5 depicts the performance measures comparison between the proposed method and state-of-the-art literature. Our proposed method has outperformed the methods by a good margin due to the efficient data preprocessing and data enhancement with data augmentation. Without performing the feature selection and data preprocessing steps, the accuracy is less than the available methods. Our proposed methodology has improved accuracy, precision, recall, and f1-score by 7.96%, 11.2%, 6.32%, and 7.97%, respectively, which is a great achievement for the Urdu community.

For a single forward pass through an LSTM layer, the time complexity is approximately $O(T*N2)$, where T is the sequence length and N is the number of LSTM units. However, during training, multiple iterations over the data are needed, so the overall complexity is higher.

The runtime complexity of an SVM primarily depends on the number of support vectors, denoted by 'nSV', and the number of input features, denoted by 'n'. For linear SVM, the training time complexity is typically $O(n * nSV)$, making it efficient for high-dimensional data but can be slow if the number of support vectors is very high.

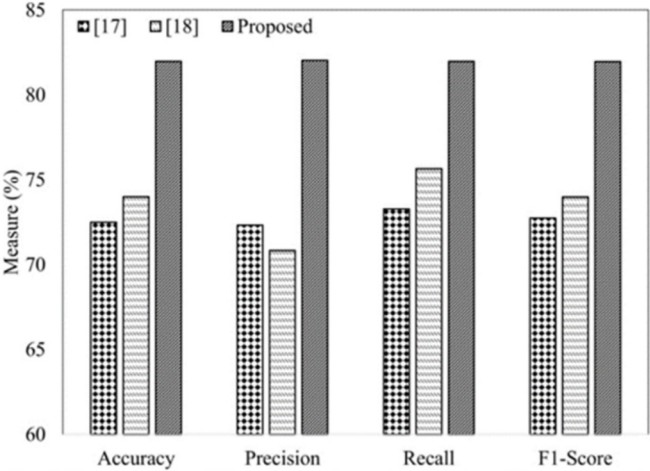

**Fig 5. Performance comparison with the literature.**

## 6. Conclusions and future directions

After performing different experiments on noisy, fewer, clean, and enhanced data, it is concluded that when data is noisy, then underfitting of the model during training occurs, and any DL model does not converge. Data augmentation can help in enhancing data and aids diversity in the data which avoids underfitting and overfitting of a model. For any automated model, data should be neat and clean. As textual data is often very noisy, it is not efficient to train a model with good performance.

Data preprocessing and cleaning textual data are essential for better performance and prevent a model's under fitting. In this research, we have achieved the best possible results for detecting threats from the Urdu language using the proposed data processing step with the help of the proposed Urdu stop words list and stemming dictionary. Due to the diverse nature of Urdu, the stemming dictionary is limited to the used dataset. This dictionary can be further extended in the future to enhance the performance of any automated model. Moreover, transfer learning-based sequential models can also be applied to efficiently preprocess data to enhance the detection rate further.

## Author Contributions

**Conceptualization:** Ashraf Ullah, Khair Ullah Khan, Aurangzeb Khan, Sheikh Tahir Bakhsh.

**Data curation:** Ashraf Ullah, Bibi Saqia.

**Formal analysis:** Ashraf Ullah, Sheikh Tahir Bakhsh, Bibi Saqia.

**Funding acquisition:** Sheikh Tahir Bakhsh.

**Investigation:** Ashraf Ullah.

**Methodology:** Ashraf Ullah.

**Project administration:** Ashraf Ullah, Sajida Akbar.

**Resources:** Ashraf Ullah, Sajida Akbar.

**Software:** Ashraf Ullah.

**Supervision:** Khair Ullah Khan, Aurangzeb Khan.

**Validation:** Ashraf Ullah, Atta Ur Rahman, Sajida Akbar.

**Visualization:** Ashraf Ullah, Atta Ur Rahman.

**Writing – original draft:** Ashraf Ullah.

**Writing – review & editing:** Ashraf Ullah.

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
