## [Decision Letter · Decision Letter 0]

18 Oct 2023

PONE-D-23-26335Threatening Language Detection from Urdu Data with Deep Sequential ModelPLOS ONE

Dear Dr. Ullah,

Thank you for submitting your manuscript to PLOS ONE. After careful consideration, we feel that it has merit but does not fully meet PLOS ONE’s publication criteria as it currently stands. Therefore, we invite you to submit a revised version of the manuscript that addresses the points raised during the review process.

We look forward to receiving your revised manuscript.

Kind regards,

M. Firoz Mridha

Academic Editor

PLOS ONE

“Unfunded studies.”

5. We note that Figure 1 in your submission contain copyrighted images. All PLOS content is published under the Creative Commons Attribution License (CC BY 4.0), which means that the manuscript, images, and Supporting Information files will be freely available online, and any third party is permitted to access, download, copy, distribute, and use these materials in any way, even commercially, with proper attribution. For more information, see our copyright guidelines: http://journals.plos.org/plosone/s/licenses-and-copyright.

Additional Editor Comments:

1. The novelty of the paper is not justified. Need to benchmark the results with other datasets and Baes line Algorithms.

2. The figure of the paper is not clear and up to the mark. Organization of the paper need to improve.

2. It is desired to discuss the model complexity of the proposed model and the baseline models. This would show their efficiency in terms of runtime and space complexity.

3. I suggest the authors give a problem statement or use case in Methodology so as to help readers understand how to make a recommendation based on the dataset.

4. I was somewhat confused about the motivations described in the introduction. I suggest the authors show the detailed process of parameters. However, the proposed method shows good performance. I also suggest the authors to adding a part of applications to discuss the accurately identifying Threatening Language Detection based the proposed method and obtained results.

Reviewers' comments:

Reviewer's Responses to Questions

**Comments to the Author**

1. Is the manuscript technically sound, and do the data support the conclusions?

Reviewer #1: Yes

Reviewer #2: Yes

2. Has the statistical analysis been performed appropriately and rigorously? 

Reviewer #1: Yes

Reviewer #2: Yes

3. Have the authors made all data underlying the findings in their manuscript fully available?

Reviewer #1: Yes

Reviewer #2: Yes

4. Is the manuscript presented in an intelligible fashion and written in standard English?

Reviewer #1: No

Reviewer #2: No

5. Review Comments to the Author

Reviewer #1: I will recommend this manuscript a minor revision to reorganize.

1. It's a well written introductory section. However, in the later section, I cannot see a similar trend.

2. There are some clear constraints in the introductory section on whether a mixed-method strategy can be used. I would also recommend that the dissertation be focused on current shortcomings and differences.

3. The scope and importance of research is not adequately clarified. The simple reader does not understand why this subject is important and what is the significance of new research results on such real-world applications, as they appear in our everyday lives.

4. There is no contrast of the latest approach introduced to other current methods in the results section (unless I have ignored).

5. Add these references in your updated manuscript:

• K. M. Hasib, M. R. Islam, S. Sakib, M. A. Akbar, I. Razzak and M. S. Alam, "Depression Detection From Social Networks Data Based on Machine Learning and Deep Learning Techniques: An Interrogative Survey," in IEEE Transactions on Computational Social Systems, vol. 10, no. 4, pp. 1568-1586, Aug. 2023, doi: 10.1109/TCSS.2023.3263128.

• Hasib, Khan Md, Nurul Akter Towhid, Kazi Omar Faruk, Jubayer Al Mahmud, and M. F. Mridha. "Strategies for enhancing the performance of news article classification in Bangla: Handling imbalance and interpretation." Engineering Applications of Artificial Intelligence 125 (2023): 106688.

• K. M. Hasib, A. Tanzim, J. Shin, K. O. Faruk, J. A. Mahmud and M. F. Mridha, "BMNet-5: A Novel Approach of Neural Network to Classify the Genre of Bengali Music Based on Audio Features," in IEEE Access, vol. 10, pp. 108545-108563, 2022, doi: 10.1109/ACCESS.2022.3213818.

Reviewer #2: The paper represents threating text detection using LSTM where data augmentation was used to increase data instances. The proposed method showed significant improvement in the result. However, the authors contributed two important resources such as vocabulary and stop word list for Urdu language.

As the authors are using sequential models like LSTM, I think the stop word removal was not required. At least, authors should see the change in the result without removing stop words.

Please rewrite some sentences in abstract and text where there is lack of clarity.

6. PLOS authors have the option to publish the peer review history of their article (what does this mean?). If published, this will include your full peer review and any attached files.

Reviewer #1: No

Reviewer #2: No

---

## [Author Response · Author response to Decision Letter 0]

15 Dec 2023

Dear Editor,

It is stated that we have thoroughly checked the editor and reviewers comments and revised the manuscript accordingly. Moreover we have provided required dataset as attachment. 

We hope that the revised manuscript will be considered for possible publication at your esteemed Journal. 

Looking forward to hear from your good self if further changes are required. 

Best Regards

Ashraf Ullah

---

## [Decision Letter · Decision Letter 1]

29 Jan 2024

PONE-D-23-26335R1Threatening Language Detection from Urdu Data with Deep Sequential ModelPLOS ONE

Dear Dr. Ullah,

Thank you for submitting your manuscript to PLOS ONE. After careful consideration, we feel that it has merit but does not fully meet PLOS ONE’s publication criteria as it currently stands. Therefore, we invite you to submit a revised version of the manuscript that addresses the points raised during the review process.

We look forward to receiving your revised manuscript.

Kind regards,

Toqir Rana

Academic Editor

PLOS ONE

Journal Requirements:

Reviewers' comments:

Reviewer's Responses to Questions

**Comments to the Author**

1. If the authors have adequately addressed your comments raised in a previous round of review and you feel that this manuscript is now acceptable for publication, you may indicate that here to bypass the “Comments to the Author” section, enter your conflict of interest statement in the “Confidential to Editor” section, and submit your "Accept" recommendation.

Reviewer #1: All comments have been addressed

Reviewer #2: All comments have been addressed

2. Is the manuscript technically sound, and do the data support the conclusions?

Reviewer #1: Yes

Reviewer #2: Yes

3. Has the statistical analysis been performed appropriately and rigorously? 

Reviewer #1: Yes

Reviewer #2: Yes

4. Have the authors made all data underlying the findings in their manuscript fully available?

Reviewer #1: Yes

Reviewer #2: Yes

5. Is the manuscript presented in an intelligible fashion and written in standard English?

Reviewer #1: Yes

Reviewer #2: Yes

6. Review Comments to the Author

Reviewer #1: I will recommend a minor revision to reorganize the paper.

1. It's a well written introductory section. However, in the later section, I cannot see a similar trend.

2. There are some clear constraints in the introductory section on whether a mixed-method strategy can be used. I would also recommend that the dissertation be focused on current shortcomings and differences.

3. The scope and importance of research is not adequately clarified. The simple reader does not understand why this subject is important and what is the significance of new research results on such real-world applications, as they appear in our everyday lives.

4. There is no contrast of the latest approach introduced to other current methods in the results section (unless I have ignored).

Reviewer #2: (No Response)

7. PLOS authors have the option to publish the peer review history of their article (what does this mean?). If published, this will include your full peer review and any attached files.

Reviewer #1: No

Reviewer #2: No

---

## [Author Response · Author response to Decision Letter 1]

2 Mar 2024

Comments and solutions

Reviewer #1: I will recommend a minor revision to reorganize the paper.

1. It's a well written introductory section. However, in the later section, I cannot see a similar trend.

Solution: In the revised draft, we have made concerted efforts to enhance clarity throughout the remaining sections. We have meticulously reviewed and either removed or rephrased any statements that were deemed irrelevant or ambiguous. Our aim was to ensure that the revised draft is more concise, coherent, and focused. We are optimistic that these revisions will address any concerns raised by the esteemed reviewer and lead to the approval of our manuscript. Thank you for your valuable feedback.

2. There are some clear constraints in the introductory section on whether a mixed-method strategy can be used. I would also recommend that the dissertation be focused on current shortcomings and differences.

Solution: In the revised version, we have incorporated an analysis of current shortcomings and differences in the introduction section. We believe that by acknowledging and addressing these challenges, our work contributes significantly to advancing the field of Urdu language processing and threat detection, thereby meeting the reviewer's expectations for a comprehensive assessment of the current landscape.

3. The scope and importance of research is not adequately clarified. The simple reader does not understand why this subject is important and what is the significance of new research results on such real-world applications, as they appear in our everyday lives.

Solution: The introduction section now explicitly outlines the scope and importance of the current work. It aims to tackle the challenges in Urdu language processing for threat detection on social media platforms. By introducing advanced preprocessing techniques and leveraging deep sequential models like Long Short-Term Memory (LSTM) units, the research endeavors to improve the accuracy of threat identification in Urdu textual data. This effort is vital for enhancing safety in online spaces by effectively mitigating threats across various social media platforms in Urdu.

4. There is no contrast of the latest approach introduced to other current methods in the results section (unless I have ignored).

Solution: Thank you for your feedback. We acknowledge the need to provide a clearer contrast between our latest approach and other current methods in the results section. Indeed, Table 6 offers a methodology comparison where we compared our approach with SVM, CNN, LR, and MLP. 

Furthermore, we appreciate your insights into future research directions. We are committed to advancing the field and addressing any remaining limitations. In our future work, we plan to explore ensemble learning techniques, incorporate domain-specific knowledge, address imbalanced data issues, and extend our approach to other languages and platforms. These avenues of investigation will enrich our understanding and contribute to the continued progress of threat detection in online environments. Thank you for bringing these points to our attention, and we will incorporate them into our revisions accordingly.

---

## [Decision Letter · Decision Letter 2]

27 Mar 2024

Threatening Language Detection from Urdu Data with Deep Sequential Model

PONE-D-23-26335R2

Dear Dr. Ullah,

We’re pleased to inform you that your manuscript has been judged scientifically suitable for publication and will be formally accepted for publication once it meets all outstanding technical requirements.

Kind regards,

Toqir Rana, Ph.D.

Academic Editor

PLOS ONE

Additional Editor Comments (optional):

Reviewers' comments:

Reviewer's Responses to Questions

**Comments to the Author**

1. If the authors have adequately addressed your comments raised in a previous round of review and you feel that this manuscript is now acceptable for publication, you may indicate that here to bypass the “Comments to the Author” section, enter your conflict of interest statement in the “Confidential to Editor” section, and submit your "Accept" recommendation.

Reviewer #1: All comments have been addressed

2. Is the manuscript technically sound, and do the data support the conclusions?

Reviewer #1: Yes

3. Has the statistical analysis been performed appropriately and rigorously? 

Reviewer #1: Yes

4. Have the authors made all data underlying the findings in their manuscript fully available?

Reviewer #1: Yes

5. Is the manuscript presented in an intelligible fashion and written in standard English?

Reviewer #1: Yes

6. Review Comments to the Author

Reviewer #1: Recommendation: Accept

However, there are some modifications are required before accepting the manuscript.

1. There are so many typos and grammatical mistakes in the manuscript. Need to proofread the manuscript again.

2. The quality of the figures should be improved for better understanding.

7. PLOS authors have the option to publish the peer review history of their article (what does this mean?). If published, this will include your full peer review and any attached files.

Reviewer #1: No

---

## [Editor Report · Acceptance letter]

30 Apr 2024

PONE-D-23-26335R2 

PLOS ONE

Dear Dr. Ullah, 

I'm pleased to inform you that your manuscript has been deemed suitable for publication in PLOS ONE. Congratulations! Your manuscript is now being handed over to our production team.

Kind regards, 

on behalf of

Dr. Toqir Rana 

Academic Editor

PLOS ONE